# Effects of a SMART Goal Setting and 12-Week Core Strength Training Intervention on Physical Fitness and Exercise Attitudes in Adolescents: A Randomized Controlled Trial

**DOI:** 10.3390/ijerph19137715

**Published:** 2022-06-23

**Authors:** Yijuan Lu, Kehong Yu, Xiaomei Gan

**Affiliations:** 1Department of Sport Science, College of Education, Zhejiang University, Hangzhou 310027, China; 11703023@zju.edu.cn (Y.L.); 11603020@zju.edu.cn (X.G.); 2Center for Sports Modernization and Development, Zhejiang University, Hangzhou 310027, China

**Keywords:** student, RCT, physical ability, attitude, core stability, target setting

## Abstract

This study aimed to analyze the impacts of a 12-week core strength training (CST) and goal-setting (GS) program on the core endurance, agility, sprinting, jumping, grip strength, and exercise attitude in a group of adolescents. This study followed a randomized parallel design in which 362 adolescents (age: 14.5 ± 1.07 years; body mass index: 19.82 ± 3.64) were allocated to a GS (*n* = 89), CST (*n* = 92), or GS + CST (*n* = 90) program or to a control group (*n* = 91). Participants were assessed two times (baseline and postintervention) for the following tests: (i) 50 m dash, (ii) grip strength, (iii) long jump, (iv) 1000 m running for boys and 800 m for girls, (v) core endurance, and (vi) exercise attitude. Significant differences (*p* < 0.05, *η*^2^*_p_* = 0.035−0.218) were found between the four groups of the six components of physical fitness and the three components of attitude toward exercise (target attitudes, behavioral habits, and sense of behavioral control). Between-group analysis revealed that the GS + CST had significant advantages (*p* < 0.05) over the CON in terms of the 50 m dash (*Cohen’s d* = 0.06), grip strength (*Cohen’s d* = 0.19_left, 0.31_right), 800/1000 m running (*Cohen’s d* = 0.41), core endurance (*Cohen’s d* = 0.95), and sense of behavioral control (*Cohen’s d* = 0.35). Between-group analysis also revealed that the CST had significant advantages over the CON in terms of grip strength (*Cohen’s d* = 0.27_left, 0.39_right), 50 m (*Cohen’s d* = 0.04), long jump (*Cohen’s d* = 0.21), 800/1000 m (*Cohen’s d* = 0.09), and core stability (*Cohen’s d* = 0.63), which were significantly different from CON (*p* < 0.05). GS differed from CON only on 50 m (*Cohen’s d* = 0.02) and core stability (*Cohen’s d* = 0.13) with a small effect (*p* < 0.05). We conclude that the combined intervention of GS and CST is more effective in promoting fitness in adolescents, i.e., GS + CST > CST and GS + CST > GS.

## 1. Introduction

The physical inactivity and declining physical health of adolescents have become a global health problem. Studies have shown that no more than 45% of students meet the recommended level of 60 min of moderate and vigorous physical activity (MVPA) [1,2], with only 8% of the adolescent population meeting the standard [3]. In Finland, 90% of girls and 77% of boys did not meet the daily recommended physical activity in their self-reports [4]. In several European countries, accelerometer measurements in children aged 10–12 years showed that only 4.6% of girls, as well as 16.8% of boys, meet the recommended amount [5].

According to nationwide surveys, the level of physical activity and health status of Chinese school-aged youths showed a downward trend from approximately 22.7% in 2010 to only approximately 8.9% in 2014 [6]. Although many policies have been put in place to improve adolescents’ PA levels, less than one-third of Chinese adolescents could meet the recommended level of 60 min of MVPA in 2016 [6]. As the physical activity level of teenagers can continue to decrease with age, which could be well above the recommended MVPA at 9 years but only 17% by 15 years [7], it is very important to conduct effective interventions to improve the physical activity of Chinese adolescents.

Schools are important places for promoting physical activity [8], especially physical education (PE) courses, which provide opportunities for students to not only participate in sports but also develop the basic motor skills, knowledge, and attitudes needed for lifelong physical activity. Because of the pressure to advance to higher education, adolescent students face heavy study and homework burdens that greatly reduce their time available for participating in extracurricular physical activities. Therefore, the physical education classroom must become an important place for motivating students to participate in moderate-to-vigorous physical activities. The importance of PE classes cannot be overlooked [9,10,11,12,13] as they involve students in nearly every age group; therefore, high-quality PE classes have a profound health impact on almost all secondary school students [14]. In a systematic review of 14 studies of interventions to increase students’ effective learning time in school physical education classes, Lonsdale et al. [11] found that students in the intervention condition spent 24% more time in PE class for moderate-to-vigorous physical activity than students in the regular practice condition. The U.S. Physical Activity Guidelines Study [15] also reported strong evidence for physical education interventions to increase student physical activity levels in class, using key strategies such as (1) developing and implementing a well-designed physical education schedule, (2) enhancing hands-on instruction with a high level of moderate-to-vigorous physical activity, and (3) providing appropriate training for teachers. Thus, it is clear that improving classroom teaching behaviors and the quality of instruction is necessary for promoting students’ physical activity levels and physical health in the classroom.

In recent decades, there has been a significant increase in physical inactivity and sedentary behavior among Chinese adolescents [16,17,18], which has been associated with a decline in physical fitness [9] and a rising trend in obesity [19]. To this end, the state has introduced a series of physical health promotion policies that have contributed to the implementation of the guiding ideology of “health first,” the strengthening of school physical education, the promotion of students’ active participation in physical activity, the development of good exercise habits, and the improvement of physical health [8,20]. Among these policy efforts, the PE examination system (PES) plays an important role in school sports, but it also produces many disadvantages brought about by “examination-based education.” PE entrance examination for senior high school (PEeeshs) is an important part of the PES. In China, every junior high school student must take a high school entrance exam for physical education and the exam is worth 30–100 points toward the total score of the junior high school entrance exam. The items of the test vary from province to province and city to city in China. In Hangzhou, Zhejiang Province, where this study was conducted, for example, the test items include endurance (1000 m/boys, 800 m/girls); strength and jumping (solid ball throw, standing long jump, pull-ups/boys, sit-ups/girls); and ball exercises: Soccer dribbling around the goal, volleyball mat, basketball dribbling and lay-ups (one of three choices). The exams are tested by the local Ministry of Education and are administered once a year, usually in April. To cope with the physical fitness test and to improve the promotion rate and complete the task of PEeeshs, many schools turn PE classes into “training classes” in the first year to cope with the PEeeshs. The main teaching task of physical education teachers is to organize students to practice and pass the PEeeshs, replacing the rich and interesting teaching content [21], which makes students less motivated to exercise in PE class and indirectly affects their motivation and interest in physical education outside of class.

The theory of planned behavior (TPB) proposed by Ajzen [22] can help us understand how people change their behavior patterns. This model suggests that the most important determinant of an individual’s behavior is their intention to perform that behavior, with three cognitive variables—attitudes, subjective norms, and perceived behavioral control—which are said to be the direct determinants of intention. Attitudes represent a key explanatory variable in many theories of health behavior; research has shown that attitudes predict both intention and behavior [23]. The TPB provides a useful conceptual framework for marking a sense of the complexities of human social behavior [24]. The attitude toward physical activity is a combination of individuals’ cognitive evaluations, affective experiences, and behavioral intentions related to physical activity [25], which are all essential psychological factors for individuals to persist in physical activity, and adolescents’ positive attitudes toward physical activity have positive impacts on their physical fitness [26].

Goal setting (GS) is a conscious and effective way to motivate individuals to be physically active and is an important factor in successful behavior change. Planning is ineffective without goals both short and long term. Goal-setting theory (GST), proposed by Locke [27,28], argues that effective goals require five principles: Goals must be clear, challenging, complex, and committed to and must also allow for feedback. Studies have confirmed the effectiveness of GS for interventions related to students’ physical activity behaviors, such as works by Spruijt-Metz (2008) and Weaver (2017), who stated that teaching students GS physical education lessons could be an effective strategy for promoting regular physical activity and increasing aerobic fitness [29,30]; Wilson (2017) used GS for children’s physical activity and enjoyment [31], and McDonald (2015) intervened with students by arranging for them to receive a SMART GS curriculum [32], both of which showed some intervention effects. Additionally, Japanese interventions for promoting students’ physical strength attached great importance to GS, not only as the intended purpose of the activity to achieve direction, motivation, and cohesion but also as a criterion for decision-making and a basis for assessment. The contribution of GST applied to interventions is that by quantifying it, exercise for students has a target reference and can serve as a form of motivation, but its disadvantage is that it is not easy to judge the rationalization of individual GS in different contexts. The SMART principle of GS is one of the ways to set goals effectively and was first proposed by management guru Peter Drucker in his book *The Practice of Management*. The SMART principle consists of five components of effective goals: They must be specific, measurable, attainable, relevant, and timely [33]. S stands for specific, which means that the goal setting or performance evaluation criteria must be specific so that people know what to do. M stands for measurable, which means that the goal or target should be measurable and able to give clear judgment, such as through data. A stands for attainable, which means that when setting goals for yourself or others, the goals should not be too high or too low; if they are too high, it is easy to discourage people; if they are too low and unchallenging, there is no need to work hard to reach them. R stands for relevant, which means that there should be some relevance between the goal and the target, regarding the overall goal or direction. T represents the time bounds, that is, the deadline for a goal; if there is no deadline, then it is essentially invalid, which is the biggest enemy of procrastination.

Core strength is a hot issue in current academic research [34,35], and many domestic and international scholars have conducted theoretical studies on the concept including the regions of the core and its effects on movement and rehabilitation [36,37,38,39,40]. Sharma et al. [41] found a significant improvement in the force of continuous and obstacle jumps after 9 months of trunk strength training, and Granacher et al. [42] found a significant improvement in lateral jump tests after a core strength intervention. Prieske [43] also observed significant improvements in trunk strength, sprinting, and other abilities after trunk strength training. Kibler [37] argued that core stability not only generates force in human movement but also carries the burden of transmitting force. Evidently, increased core stability achieved through core strength exercises can promote the development of other physical qualities. Past studies have also reported encouraging results through modified physical education curricula, physical education teacher training, after-school physical education interventions, and increased involvement of policymakers and parents. In China, research on core strength exercises has mainly focused on the field of competitive sports training, and there is little research on school sports that addresses student physical health promotion. To this end, for this study, we designed an interesting core strength training (CST) program for the physical exercise component of one school system’s curriculum, replacing the original monotonous training content on the physical fitness test.

Based on the above, this study (1) proposes a comprehensive intervention theoretical framework based on TPB and GST, (2) designs an intervention program based on the comprehensive intervention theoretical framework, and (3) aims to evaluate the effect of the intervention on enhancing adolescents’ exercise attitudes and physical fitness.

## 2. Materials and Methods

### 2.1. Study Design

This study was based on the TPB and GST as frameworks for designing the intervention program (shown in Figure 1). Based on TPB, our study optimizes the content of physical fitness exercises in the physical education classroom, proposes a core strength training (CST) program to improve students’ motivation and fitness, and gives students quantitative goal-setting (GS) based on GST to motivate them to exercise. Four groups were included: (1) Experimental group 1 (GS), which performed a goal-setting intervention program; (2) experimental group 2 (CST), which performed a core-strength training intervention program; (3) experimental group 3, which underwent a combined program (GS + CST); and (4) the control group (CON), in which participants were asked to maintain their usual routines.

This 12-week RCT followed a single-blinded design and was performed between October 2020 and January 2021 (shown in Figure 2). This study was designed as a randomized controlled trial. Before the trial was initiated, the participants were randomly assigned to one of the four groups using a computer-generated simple randomization procedure [44]. This RCT was reported according to the Consolidated Standard of Reporting Trials (CONSORT 2010) guidelines (http://www.consort-statement.org (accessed on 15 June 2022)). In addition, a concise overview of the intervention programs was described according to the CONSORT 2010 checklist (http://www.consort-statement.org (accessed on 15 June 2022)). This study lasted 15 weeks, during which the active intervention was 12 weeks, while recruitment, screening, and pre- and post-measurement took up the remaining 3 weeks. In detail, the first week was used to screen for and recruit suitable subjects. Then, during the second week, all students finished the questionnaire and fitness test. Finally, in week 15, the post-test procedure was identical to the baseline procedure after the 12-week active intervention period. The CST and CST + GS intervention lasted for 12 weeks, three times a week for 7–10 min each time. The GS intervention took place once in the first week. All the participants were tested two times: (a) Before the CST and GS and (b) 12 weeks after the CST.

### 2.2. Participants

Three hundred and sixty-eight students were chosen from eight classes in two middle schools in Hangzhou, Zhejiang Province, China, by random sampling, based on their voluntary participation in this study. The inclusion criteria required that (a) students were able to perform daily physical education classes, and (b) students did not participate in the school’s after-school sports club. Four students were excluded because they participated in daily training with the school’s soccer club. Of the 368 candidates, 364 volunteers met the inclusion criteria, as described in Figure 3. Before the study, the demographic information of these students was collected, then we allocated them to the intervention (GS, CST, GS + CST) and control groups. A total of 364 students were pretested at baseline, and two students (one in the GS + CST group due to a sprained ankle; one in the control group due to transfer to another school) were lost to follow-up at the posttest. All the participants completed the psychological questionnaire.

Hence, a total of 271 students of the intervention group (138 boys and 133 girls) and 91 subjects of the control group (44 boys and 47 girls) were analyzed via screening and cleaning data for questionnaires and the physical fitness test (shown in Table 1). The students’ mean ± SD age and body mass index (BMI) were 14.5 ± 1.07 years and 19.82 ± 3.64, respectively. All the students and families were fully informed about the possible problems related to the experimental procedures. The study procedures were approved by the research ethics board of Zhejiang University (No.2020-002, 22 July 2020). All the participants gave written informed consent. The analyses were processed using the SPSS 25.0 program, except for effect size calculations (*Cohen’s d*), which were processed using a web-based effect-size calculator (Effect Size Calculator (https://www.campbellcollaboration.org/ (accessed on 20 May 2022)). This calculator is a companion to the 2001 book by Mark W. Lipsey and David B. Wilson, Practical Meta-analysis, published by Sage.

### 2.3. Intervention

Experimental group 1 (GS) had one intervention in the first week and was assigned to fill out the SMART GS card (shown in Figure 4) during the first week of the intervention. We designed goal questions for each item of the physical fitness test based on the SMART principle. With the assistance of the physical education teacher, the GS groups of students filled out a questionnaire at the beginning of the first physical education period of the first week of the intervention.

Experimental group 2 (CST) performed a core strength training intervention program. CST consisted of two themes, i.e., bobby jumps and agility ladder assistance exercises, with a total of nine exercises (shown in Table 2). The CST was performed for approximately 7–10 min three times per week for 12 weeks. Table 3 shows the phases of the CST program. PE classes for Chinese adolescents last a total of 40 min and consist of three parts: The first is the warm-up part (approximately 7 min), the second is the teaching and practice part (approximately 30 min), and the third is the relaxation part (approximately 3 min). The SCT intervention took place in the second half of the second part. The active intervention and measurements were mainly carried out by two trained physical education teachers (M.G. and M.F.) from the experimental schools. The intervention was conducted by one teacher in grade 7 and one in grade 8. The pre-intervention researcher first instructed and trained the physical education teachers and provided eight sets of agility ladder equipment to the intervention schools.

Experimental group 3 underwent a combined program (GS + CST), and the control group (CON) participants were asked to maintain their daily life routines.

### 2.4. Measurements

#### 2.4.1. Anthropometric Measures

Demographic indicators, including birth year, height, and weight, were obtained from the laboratory school medical office staff.

#### 2.4.2. Physical Fitness Test

With the assistance of the experiential schools, the physical fitness data were obtained during the first week before the intervention, in strict accordance with the requirements of the Chinese National Student Physical Fitness Test. Physical fitness tests were completed by three trained experimenters assisting the physical education teachers. All tests (5 items) were completed in one physical education classroom session. Before the test, students completed a 5-minute warm-up exercise led by the physical education teacher. The tests were arranged in the order of grip strength, long jump, 50-m dash, core endurance test, and 800-/1000-m run. Since the promulgation and implementation of the National Physical Fitness Standards for Students in China in 2002, schools have been conducting tests of the Standards every school year covering all grades of students in the school, and every student knows the requirements and precautions for the tests, so teachers do not need a detailed introduction.

***Grip Strength*** The subject holds a grip strength device with the pointer facing outward and adjusts it according to the size of the palm so that the second joint of the index finger is close to a right angle, and the measurement is taken. The test subject’s body is straight, and the feet are naturally separated. The grip should touch the body or one’s clothes as little as possible and should not be swung back and forth during the measurement; it should be kept as still as possible for the test. The measurements are taken in the order of right–left, right–left two times for each hand.

***Long Jump*** Each person jumps twice, and the measurement of the longer of the two is used as the result. The test is taken barefoot or with rubber shoes but no shoes with spikes or rubber soles. When jumping, a toe step on the line (including a toe step caused by padding, continuous jumping, and other actions) is a foul, and the score is 0. A student is also given zero points for landing backward into the invalid test area.

***50 m Dash*** Subjects line up at the starting point in groups of five, and all run on hearing the “ready, run” command. The test is over when the body reaches the vertical surface of the finish line. It is considered a foul to push others, and the race will need to be re-run.

***800/1000 m Running*** Before the test, the teachers mobilized the students for a full warm-up to mentally prepare them to do their best to complete the test. For students in poor physical condition, the test duration was shortened.

***Core Endurance*** To date, no test method can accurately and comprehensively reflect core stability, and the lack of a gold standard has become an obstacle in research. Some researchers have suggested that such a standard is not possible because of the item-specific nature of core stability and that the choice of test should be based on the characteristics of the item. Waldelm and Li [45] summarized the 34 most commonly used core stability testing methods based on previous studies and classified them into five categories: Core strength, core endurance, core flexibility, core motor control, and core functional testing. Leitz [46] selected one test from each of the five categories, choosing trunk flexion and extension, dominant leg standing, dominant leg hop, sit-ups, and extensor endurance to comprehensively reflect core strength. For this study, we used a test designed by Brian McKenzie, Senior Athletics Coach (UKA4) of UK Athletics, the national governing body for athletics in the UK. Three trained experimenters accompanying the physical education teachers completed the core endurance test. Prior to the test, each tester was assigned a pen, a physical fitness test sheet (including core strength), and a clock. The testers demonstrated and explained the eight test movements and requirements.

The content and order of the tests are as follows:Participants will start in a planking position and hold for 60 s (1 point for completion).Lift their right arm off the ground and hold for 15 s (3 points for completion).Return their right arm to the ground and lift the left arm off the ground and hold for 15 s (5 points for completion).Return their left arm to the ground and lift their right leg off the ground and hold for 15 s (6 points for completion).Return their right leg to the ground and lift their left leg off the ground and hold for 15 s (10 points for completion).Lift their left leg and right arm off the ground hold for 15 s (15 points for completion.Return their left leg and right arm to the ground. Lift their right leg and left arm off the ground hold for 15 s (25 points for completion).Return to the plank exercise position (elbows on the ground) and hold this position for 30 s (35 points for completion).

It is important to note that the trunk is always in a neutral position throughout the test, and the test must be performed continuously from the first step to the eighth step. If a step in the test fails to meet the requirements, the test is over, and the total score obtained at this point is the test result. The higher the score, the better the core strength and stability.

#### 2.4.3. Physical Attitude Test

With the assistance of a physical education teacher, the researcher collected data on participants’ attitudes toward exercise during the first week before the intervention. For this study, we used the Exercise Attitude Scale [47] developed by Mao Rong-Jian to measure the students’ attitudes toward physical activity. This scale has been included in the *Handbook of Evaluation of Commonly Used Scales in Sports Science*, edited by Tension and Mao, and has often been adopted by domestic researchers to collect physical activity attitudes of primary and secondary school students and college students [25,48,49,50,51,52,53].

The scale contains the following domains: Behavioral attitudes (eight questions), target attitudes (12 questions), behavioral perceptions (seven questions), behavioral habits (10 questions), behavioral intention (eight questions), emotional experience (10 questions), sense of behavioral control (eight questions), and subjective standards (seven questions). Cronbach’s alphas for the eight subscales were 0.83, 087, 0.73, 0.89, 0.84, 0.86, 0.80, and 0.64, respectively, and the results of the total scale structure model test were *x*^2^/df = 3.67, NNFI = 0.93, CFI = 0.94, AGFI = 0.87, and RMSEA = 0.06, respectively, indicating that the scale has good structural validity.

### 2.5. Statistical Procedures

We present descriptive statistics such as M, SD, and percentage where appropriate. Chi-square statistics were used to compare the number of children in different groups based on sex. The assumptions of the ANOVA were assessed to be satisfied based on the results of the Shapiro-Wilk and Levene tests. When the data were not normally distributed, the Mann–Whitney U test was performed for between-group comparisons and the Wilcoxon matched-pair test was used for within-group comparisons. According to the Shapiro–Wilk and the Levene test results, ANCOVA assumptions were met. Factorial univariate analysis of variance (ANOVA) was used to compare baseline age, grade, gender, and body mass index (BMI) among all groups and were corrected using Bonferroni adjustments when needed. Cronbach’s alpha was assessed to determine the internal consistency for each subscale of the Exercise Attitude Scale.

A factorial univariate analysis of covariance (ANCOVA) utilizing the baseline score and other key confounders as covariates (age and BMI) was used to determine the effects of the intervention. The within-subject factor was time (two assessment points: Baseline, immediately after the intervention) and the between-subject factor was experimental condition (experimental vs. control). ANCOVA, which utilizes a baseline score as a covariate, is recommended because it increases statistical power and precision [54]. Analyses of simple effects and post-hoc Bonferroni adjustments were performed after significant interaction effects by overall ANCOVA were confirmed. The calculated effect size was *η*^2^*_p_*, with effect sizes rated as follows: Small, 0.01 ≤ *η*^2^*_p_* < 0.06; medium, 0.06 ≤ *η*^2^*_p_* < 0.14; or large, *η*^2^*_p_* ≤ 0.14 [55]. We set statistical significance at *p* < 0.05 for all tests. The effect sizes for mean differences were expressed as *Cohen’s d* (difference in means divided by the standard deviation of the difference) with values of 0.2, 0.5, and 0.8 denoting small, medium, and large effect sizes, respectively [56].

## 3. Results

### 3.1. Effects of Intervention on Students’ Physical Fitness

An ANOVA was used to test for differences between groups for the baseline test. The results showed that except for the significant difference in right-hand grip strength (*p* = 0.03), the differences in the rest of the physical fitness tests were not significant. Gender differences between groups were analyzed using the chi-square test, and *p*-values were corrected using the Bonferroni method. The results showed that the gender differences between the four groups were not significant (*x*^2^ = 0.273, *p* = 0.965).

The intra-group comparison for the physical fitness component is presented in Table 4. The intervention groups presented a significant increase in both left and right handgrip tests (GS: *Cohen’s d* = 0.09 and 0.15; CST: *Cohen’s d* = 0.72 and 0.81; GS + CST: *Cohen’s d* = 0.17 and 0.24). For 50 m and 800/1000 m, the intervention group showed a significant increase in speed (GS: *Cohen’s d* = 0.22 and 0.17; CST: *Cohen’s d* = 0.4 and 0.55; GS + CST: *Cohen’s d* = 0.32 and 0.56). Differences between groups also showed that the intervention group significantly improved in the long jump and core endurance tests (GS: *Cohen’s d* = 0.25 and 0.21; CST: *Cohen’s d* = 0.51 and 0.49; GS + CST: *Cohen’s d* = 0.35 and 0.63). Additionally, within-group changes in the control group were also significant for the long jump (*p* < 0.001, *Cohen’s d* = 0.27) and 1000 m/800 m (*p* < 0.001, *Cohen’s d* = 0.25).

The rapid physical and mental developments during adolescence are characterized by significant gender differences. We included gender as a factor because of the significant differences (*p* < 0.001) in the preintervention scores on six of the physical fitness test items (excluding the 50 m dash, *p* = 0.189).

After controlling for age, BMI, and baseline scores, the results of the ANCOVAs showed a significant group effect for the physical fitness component (left-hand grip strength: *F* = 22.956, *p* < 0.001, *η*^2^*_p_* = 0.167; right-hand grip strength: *F* = 31.888, *p* < 0.001, *η*^2^*_p_* = 0.218; 50 m: *F* = 9.5, *p* < 0.001, *η*^2^*_p_* = 0.079; long jump: *F* = 4.041, *p* = 0.008, *η*^2^*_p_* = 0.035; 800/1000 m: *F* = 10.181, *p* < 0.001, *η*^2^*_p_* = 0.084; core endurance: *F* = 21.046, *p* < 0.001, *η*^2^*_p_* = 0.157). We found no significant interactions between groups or for gender in the mixed ANCOVA for the 50 m dash (*p* = 0.083, *η*^2^*_p_* = 0.020), grip strength (*p* = 0.417/0.317, *η*^2^*_p_* = 0.008/0.01), long jump (*p* = 0.969, *η*^2^*_p_* = 0.01), 1000 m/800 m (*p* = 0.724, *η*^2^*_p_* = 0.004), and core endurance (*p* = 0.076, *η*^2^*_p_* = 0.020). Table 5 present the results from between-group analyses.

Variables with significant differences were compared with the Bonferroni post-hoc to further understand the differences between the four groups. For left-hand grip strength performance, the CST was higher than the GS group by 2.97 kg (95%CI: 2.12–3.83, *p* < 0.001, *Cohen’s d* = 0.05), higher than the CST + GS by 2.31 kg (95%CI: 1.45–3.18, *p* < 0.001, *Cohen’s d* = 0.32), and higher than the control group by 3.24 kg (95%CI: 2.39–4.09, *p* < 0.001, *Cohen’s d* = 0.27). CST + GS was higher than the control group by 0.93 kg (95%CI: 0.08–1.78, *p* < 0.05, *Cohen’s d* = 0.19). For right-hand grip strength performance, CST was higher than the GS group by 3.54 kg (95%CI: 2.61–4.46, *p* < 0.001, *Cohen’s d* = 0.43), higher than the CST + GS by 2.56 kg (95%CI: 1.63–3.50, *p* < 0.001, *Cohen’s d* = 0.06), and higher than the control group by 4.27 kg (95%CI: 3.35–5.19, *p* < 0.001, *Cohen’s d* = 0.39). For long jump performance, the CST was 5.01 cm (95%CI: 1.65–8.38, *p* < 0.01, *Cohen’s d* = 0.21) higher than the GS group and 5.17 cm (95%CI: 1.84–8.50, *p* < 0.01, *Cohen’s d* = 0.29) higher than the control group. For 50 m dash performance, CST was faster than GS by 0.14 s (95%CI: 0.04–0.25, *p* < 0.01, *Cohen’s d* = 0.25) and faster than control group by 0.26 s (95%CI: 0.16–0.36, *p* < 0.001, *Cohen’s d* = 0.04); GS was faster than the control group by 0.12 s (95%CI: 0.01–0.22, *p* < 0.05, *Cohen’s d* = 0.02); CST + GS was faster than the control group by 0.21 s (95%CI: 0.10–0.31, *p* < 0.001, *Cohen’s d* = 0.06). For 800/1000 m running performance, the CST speed was 0.24 min faster than GS (95%CI: 0.12–0.35, *p* < 0.001, *Cohen’s d* = 0.25) and 0.13 min faster than the control group (95%CI: 0.01–0.24, *p* < 0.001, *Cohen’s d* = 0.09); CST + GS was 0.29 min faster than GS (95%CI: 0.18–0.40, *p* < 0.001, *Cohen’s d* = 0.58); CST + GS was 0.18 min faster than the control group (95%CI: 0.07–0.29, *p* < 0.01, *Cohen’s d* = 0.41). For core strength scores, CST was higher than GS by 8.13 points (95%CI: 0.57–15.70, *p* < 0.05), higher than the control group by 15.88 points (95%CI: 8.37–23.38, *p* < 0.001); CST + GS was higher than CST by 13.14 points (95%CI: 5.34–20.75, *p* < 0.01), higher than GS by 21.27 points (95%CI: 13.65–28.9, *p* < 0.001), and higher than the control group by 20.02 points (95%CI: 21.52–36.52, *p* < 0.001). Figure 5 shows the results of the post-hoc test between GS, CST, GS + CST, and CON.

### 3.2. Effects of Intervention on Students’ Exercise Attitude

ANOVA was used to test for between-group differences in the baseline tests. The results showed that there were significant differences in the exercise attitude component, except for target attitudes and behavioral perceptions (*p* > 0.05). 

Table 6 present the descriptive statistics for the students’ within-group preintervention and postintervention attitudes toward exercise. The results of within-group differences in students’ attitudes toward exercise indicators showed no significant changes in any of the eight dimensions of exercise attitudes among students in CST, GS + CST, or CON, but GS showed a significant decrease in two indicators: Target attitude (*p* = 0.028, *Cohen’s d* = 0.23) and behavioral habits (*p* = 0.04, *Cohen’s d* = 0.16).

Table 7 presents the descriptive statistics of the students’ pre- and post-intervention intergroup attitudes toward exercise. After adjustment for covariates, behavioral habits (*F* = 3.668, *p* = 0.013, *η*^2^*_p_* = 0.031), behavioral intentions (*F* = 3.354, *p* = 0.019, *η*^2^*_p_* = 0.028), and sense of behavioral control (*F* = 3.802, *p* = 0.011, *η*^2^*_p_* = 0.032) showed statistically significant differences between groups. We found no significant mixed ANCOVA interactions between groups or gender for behavioral attitudes, target attitudes, behavioral perceptions, behavioral habits, behavioral intention, emotional experience, sense of behavioral control, or subjective standards. 

Variables with significant differences were compared with Bonferroni post-hoc to further understand the differences between the four groups. For behavioral habits, the CST + GS was higher than the CST by 2.14 points (95%CI: 0.04–4.24, *p* < 0.05, *Cohen’s d* = 0.54) and higher than the GS group by 3.47 points (95%CI: 01.40–5.55, *p* < 0.05, *Cohen’s d* = 0.62). For behavioral intention: scores were 2.93 higher in the CST + GS group than in the GS (95%CI: 1.11–4.75, *p* < 0.05, *Cohen’s d* = 0.69). For sense of behavioral control, the CST + GS group was higher than the GS group by 2.8 points (95%CI: 1.0–4.6, *p* < 0.05, *Cohen’s d* = 0.63) and higher than the control group by 2.07 points (95%CI: 0.29–3.84, *p* < 0.05, *Cohen’s d* = 0.35); the CST group was higher than the GS group by 2.0 points (95%CI: 0.21–3.78, *p* < 0.05, *Cohen’s d* = 0.27). Figure 6 shows the results of the post-hoc test between GS, CST, GS + CST, and CON.

## 4. Discussion

Our findings indicate that our intervention was more effective when we combined the GS intervention with the CST intervention in the physical education classroom, i.e., GS + CST > CST and GS + CST > GS. Influenced by many factors such as the pressure of entrance exams and physical exercise habits, students’ daily time spent in moderate-to-vigorous physical activity decreases significantly after the transition from elementary to junior high school, and schools must play an important role in promoting physical exercise among junior high school students. PE class is the basic organizational form of school physical education, and they are important for students for improving their physical fitness, motor skills, and overall health and for developing lifelong awareness of physical education [57]. However, some schools have turned PE classes into boring preparation activities for the PE entrance examination; students find this “teaching to the test” dull, and it has shown negative effects such as decreasing students’ motivation to exercise and failing to develop students’ techniques and skills, especially for some low-intensity sports. Physical education teachers need to find a balance between ensuring the completion of classroom tasks and improving students’ physical fitness. The results of the effectiveness analysis of this study’s intervention showed that interspersing short periods of fun core strength combination exercises with the aid of students’ favorite agility ladders in the physical education classroom improved their physical fitness and promoted their physical activity behaviors, behavioral intentions, and sense of behavioral control.

Existing studies have found that CST has an enhancing effect on balance [58,59], strength [60], specific speed qualities [61], and dynamic postural control [62] and that core strength interventions have positive effects on sports performance and promoting athletic ability. After reviewing 16 experimental research papers on the effects of core strength on athletic ability, Fu et al. found that CST was able to positively affect the basic athletic abilities of running, jumping, throwing, swimming, and rowing relative to general training, with greater effects on running and jumping than on throwing, swimming, and rowing [63]. Additionally, after reviewing 44 experimental research papers on the effects of CST on athletic performance, this research team found positive effects of improved running, jumping, and distal speed category performance, especially for stability; we also found that the number of training weeks was positively correlated with the amount of effect [64]. A literature review of 34 randomized controlled trials of core strength interventions by Niu et al. (2018) found that CST has important value in improving muscle control, maintaining overall body posture, playing a transmission role in the core region in the kinetic chain, and improving core stability in athletes [65]. Electromyographic studies have shown that enhanced core strength can reduce the discharge of the posterior musculature of the swing leg during high-speed running vacations, so that the leg muscles can be relaxed, reducing energy expenditure during the short vacating time to prepare for the next contraction to generate more force [66]. The results of this study are consistent with those from existing studies that show that CST with short-duration single exercises and a cumulative frequency of multiple reps has a beneficial effect on the physical fitness levels of junior high school students.

The positive effects of the GS-based intervention in this study on middle school students’ physical fitness and physical activity attitudes were not as significant as the effects of the physical education classroom intervention and the combination intervention, but they were significantly higher than the findings for the control group. The results are also consistent with the results of the original trials of behavioral interventions through GS, such as GS interventions for adolescent nutrition education [67]; adolescent daily step improvement [11,68]; and aerobic fitness for students in grades 6–8 [32], all of which had significant positive effects. A systematic review by Epton et al. [69] of 141 papers on behavioral interventions through GS found that GS has positive intervention effects, is an effective approach to behavior change, and can be considered an essential component of conducting successful interventions. Desmond et al. [70], after a systematic review of 45 experimental literature on interventions for physical activity behaviors through multicomponent GS, noted that GS interventions related to physical activity behaviors had moderate positive effects (*Cohen’s d* = 0.552). Short-term goals are the most likely to have an immediate motivational impact on human action, and clear, specific, measurable goals generate a greater motivational drive and lead to good grades [71,72]. Educating students about GS is a viable and potentially effective strategy for promoting increased physical activity and physical health promotion.

## 5. Conclusions

The present parallel study design shows that CST in a PE class provides concentrated benefits in core endurance that students in other physical fitness groups did not show. CST should be considered an element to be introduced in comprehensive strength training. The use of fun combinations of exercises can be beneficial in increasing students’ motivation to practice in PE class. Assisting students with GS for exercise did not have a significant effect on improving students’ attitudes toward exercise; however, combining GS with core strength exercise interventions had a significant effect on improving students’ physical fitness that was greater than that for the core strength interventions alone. As for the scientific evidence and practical implications, we conclude that CST combined with GS has significant advantages for improving students’ physical fitness.

## Figures and Tables

**Figure 1 ijerph-19-07715-f001:**
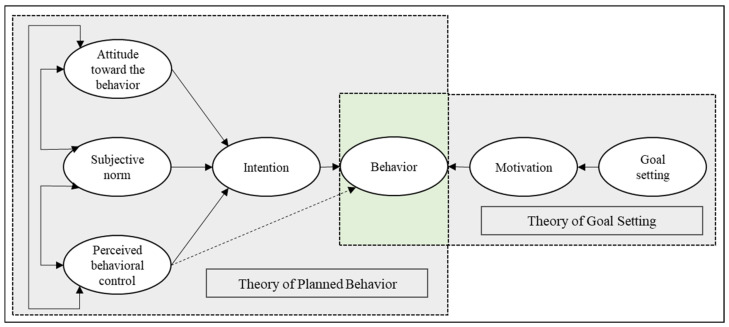
Theoretical framework.

**Figure 2 ijerph-19-07715-f002:**
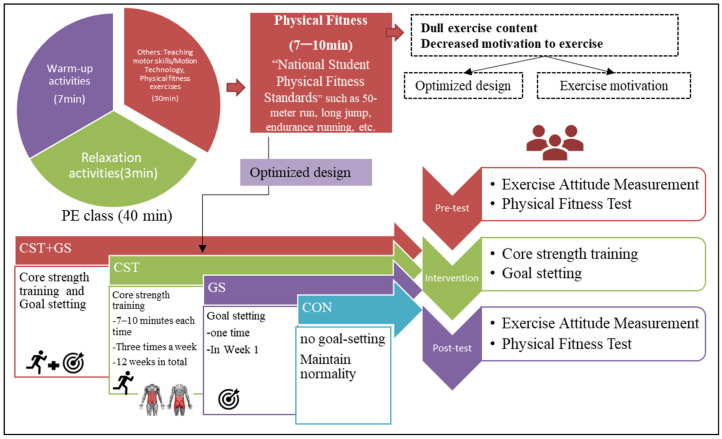
Research design flow chart.

**Figure 3 ijerph-19-07715-f003:**
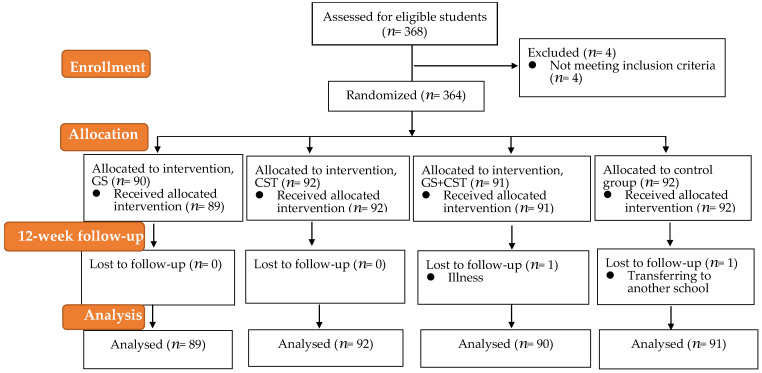
Flow diagram of the study participants.

**Figure 4 ijerph-19-07715-f004:**
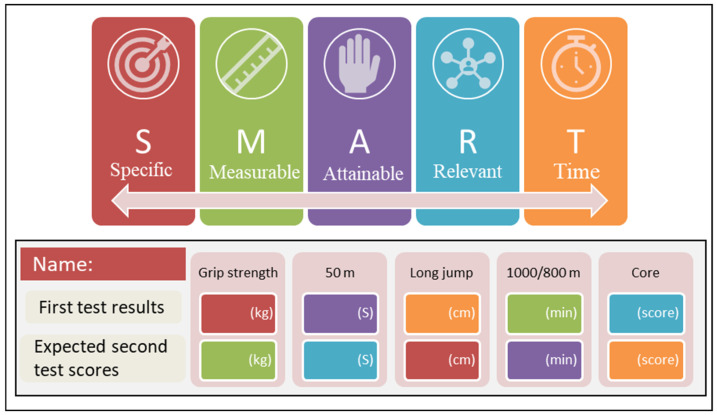
SMART goal-setting card.

**Figure 5 ijerph-19-07715-f005:**
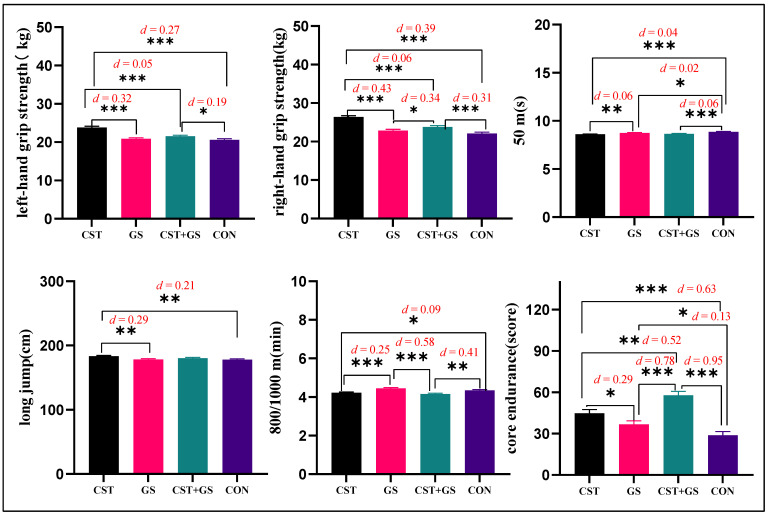
Descriptive statistics, between-group variations of means of physical fitness. GS: Goal setting intervention group; CST: Core strength training intervention group; GS + CST: Goal setting and core strength training cointervention group; CON: Control group; “*”: *p* < 0.05; “**”: *p* < 0.01; “***”: *p* < 0.001.

**Figure 6 ijerph-19-07715-f006:**
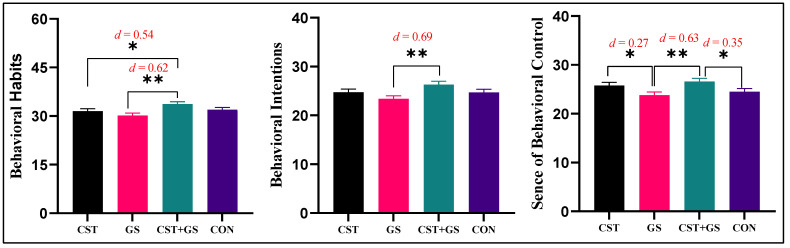
Descriptive statistics, between-group variations of means of exercise attitude (behavioral habits, behavioral intention, sense of behavioral control). GS: Goal setting intervention group; CST: Core strength training intervention group; GS + CST: Goal setting and core strength training cointervention group; CON: Control group; “*”: *p* < 0.05; “**”: *p* < 0.01.

**Table 1 ijerph-19-07715-t001:** Participant characteristics.

	GS	CST	GS + CST	CON	Total	*p*
	*n* = 89	*n* = 92	*n* = 90	*n* = 91	*n* = 362	
Grade						
7	43 (48.3%)	46 (50)	45 (50)	44 (48.4)	173	0.992
8	46 (51.7)	46 (50)	45 (50)	47 (51.6)	189	
Gender						
Boys	45 (50.6)	48 (52.2)	45 (50)	44 (48.4)	182	0.965
Girls	44 (49.4)	44 (47.8)	45 (50)	47 (51.6)	180	
BMI (kg/m^2^)						
BMI, Mean (SD)	20.37 (4.21)	19.47 (3.19)	19.57 (3.22)	19.86 (3.86)	19.82 (3.64)	0.346
<18.5	38 (42.7)	43 (46.7)	39 (43.3)	41 (45.1)		
18.5 ≤ BMI < 24	32 (36)	39 (42.4)	43 (47.8)	37 (40.7)		
≥24	19 (21.3)	10 (10.9)	8 (8.9)	13 (14.3)		
Age (yr)						
Age, Mean (SD)	14.54 (1.07)	14.59 (1)	14.42 (1.04)	14.46 (1.78)	14.5 (1.07)	0.727
13	19 (21.3)	13 (14.1)	22 (24.4)	28 (30.8)		
14	23 (25.8)	33 (35.9)	23 (25.6)	16 (17.6)		
15	27 (30.3)	26 (28.3)	30 (33.3)	24 (26.4)		
16	20 (22.5)	19 (20.7)	15 (16.7)	23 (25.3)		
17	0 (0)	1 (1.1)	0 (0)	0 (0)		

GS: Goal setting intervention group; CST: Core strength training intervention group; GS + CST: Goal setting and core strength training cointervention group; CON: Control group.

**Table 2 ijerph-19-07715-t002:** Core strength training content design.

Theme	Core Strength Training Program Details
Bobby jump	Squatting Down + Leg Thrust + Forward Jump + Vertical Jump Squatting Down + Leg Thrust + Push-Up + Forward Jump + Vertical Jump Squatting Down + Leg Thrust + Prone lunge jump + Forward Jump + Vertical Jump Squatting Down + Leg Thrust + Prone open and closed jump + Forward Jump + Vertical Jump
Agile ladder	e. Lateral climbing f. High bar leg run g. Open and close jump h. Mimic spider crawling i. Left and right lateral alternating feet jump

**Table 3 ijerph-19-07715-t003:** Phases of the CST program.

Week	PE 1	PE 2	PE 3
1	[(a × 15 REP) + (f × 5 REP) + (g × 5 REP)] × 3 REP	[(a × 15 REP) + (f × 5 REP) + (g × 5 REP)] × 3 REP	[(a × 15 REP) + (f × 5 REP) + (g × 5 REP)] × 3 REP
2	[(b × 15 REP) + (f × 5 REP) + (e × 5 REP)] × 3 REP	[(b × 15 REP) + (f × 5 REP) + (e × 5 REP)] × 3 REP	[(b × 15 REP) + (f × 5 REP) + (e × 5 REP)] × 3 REP
3	[(c × 15 REP) + (g × 5 REP) + (h × 5 REP)] × 3 REP	[(c × 15 REP) + (g × 5 REP) + (h × 5 REP)] × 3 REP	[(c × 15 REP) + (g × 5 REP) + (h × 5 REP)] × 3 REP
4	[(d × 15 REP) + (e × 5 REP) + (i × 5 REP)] × 3 REP	[(d × 15 REP) + (e × 5 REP) + (i × 5 REP)] × 3 REP	[(d × 15 REP) + (e × 5 REP) + (i × 5 REP)] × 3 REP
5	[(a × 15 REP) + (f × 5 REP) + (g × 5 REP)] × 5 REP	[(a × 15 REP) + (f × 5 REP) + (g × 5 REP)] × 5 REP	[(a × 15 REP) + (f × 5 REP) + (g × 5 REP)] × 5 REP
6	[(b × 15 REP) + (f × 5 REP) + (e × 5 REP)] × 5 REP	[(b × 15 REP) + (f × 5 REP) + (e × 5 REP)] × 5 REP	[(b × 15 REP) + (f × 5 REP) + (e × 5 REP)] × 5 REP
7	[(c × 15 REP) + (g × 5 REP) + (h × 5 REP)] × 5 REP	[(c × 15 REP) + (g × 5 REP) + (h × 5 REP)] × 5 REP	[(c × 15 REP) + (g × 5 REP) + (h × 5 REP)] × 5 REP
8	[(d × 15 REP) + (e × 5 REP) + (i × 5 REP)] × 5 REP	[(d × 15 REP) + (e × 5 REP) + (i × 5 REP)] × 5 REP	[(d × 15 REP) + (e × 5 REP) + (i × 5 REP)] × 5 REP
9	[(a × 15 REP) + (f × 5 REP) + (g × 5 REP)] × 6 REP	[(a × 15 REP) + (f × 5 REP) + (g × 5 REP)] × 6 REP	[(a × 15 REP) + (f × 5 REP) + (g × 5 REP)] × 6 REP
10	[(b × 15 REP) + (f × 5 REP) + (e × 5 REP)] × 6 REP	[(b × 15 REP) + (f × 5 REP) + (e × 5 REP)] ×6 REP	[(b × 15 REP) + (f × 5 REP) + (e × 5 REP)] × 6 REP
11	[(c × 15 REP) + (g × 5 REP) + (h × 5 REP)] × 6 REP	[(c × 15 REP) + (g × 5 REP) + (h × 5 REP)] × 6 REP	[(c × 15 REP) + (g × 5 REP) + (h × 5 REP)] × 6 REP
12	[(d × 15 REP) + (e × 5 REP) + (i × 5 REP)] × 6 REP	[(d × 15 REP) + (e × 5 REP) + (i × 5 REP)] × 6 REP	[(d × 15 REP) + (e × 5 REP) + (i × 5 REP)] × 6 REP

REP: Repetitions; PE: Physical education.

**Table 4 ijerph-19-07715-t004:** Descriptive statistics of physical fitness variables (mean, standard deviation); within-group analysis.

Variables	GS			CST			GS + CST			CON		
	Pre	Post	*d*	Pre	Post	*d*	Pre	Post	*d*	Pre	Post	*d*
Left-hand grip strength (kg)	20.1, 6.2	20.7, 6.5	0.09 *	18.7, 5.5	22.7, 5.6	0.72 ***	21.1, 6.7	22.3, 7.3	0.17 ***	20.7, 6.3	20.9, 6.7	0.03
Right-hand grip strength (kg)	21.3, 7.0	22.4, 6.8	0.15 **	20.4, 5.9	25.2, 6.18	0.81 ***	23.1, 6.7	24.7, 7.1	0.24 ***	22.5, 6.9	22.6, 6.8	0.01
Long jump (cm)	168.5, 27.9	175.6, 28.6	0.25 ***	170.5, 25.3	183.5, 25.8	0.51 ***	172.9, 26.7	182.3, 26.5	0.35 ***	171.0, 25.0	178.1, 27.0	0.27 ***
50 m (s)	8.9, 1.0	8.7, 0.9	0.22 ***	9.0, 0.9	8.7, 0.9	0.40 ***	9.1, 0.9	8.8, 0.9	0.32 ***	8.8, 0.8	8.7, 0.8	0.05
800/1000 m (min)	4.7, 0.9	4.5, 1.0	0.17 **	4.7, 0.9	4.3, 0.8	0.55 ***	4.4, 0.7	4.0, 0.7	0.56 ***	4.6, 1.0	4.4, 1.0	0.25 ***
Core endurance (score)	26.7, 30.98	33.7, 35	0.21 **	27.7, 31.4	43.6, 33.8	0.49 ***	38.5, 38.7	61.9, 35.1	0.63 ***	32.5, 38.0	29.3, 31.5	0.09

GS: Goal setting intervention group; CST: Core strength training intervention group; GS + CST: Goal setting and core strength training cointervention group; CON: Control group; statistical significance was set to *p* < 0.05; “*”: *p* < 0.05; “**”: *p* < 0.01; “***”: *p* < 0.001.

**Table 5 ijerph-19-07715-t005:** Analysis of covariance of physical fitness test indexes between experimental and control groups.

Variables	Group		Gender		Interaction Effects		Pairwise Comparison
	*F*	*η* ^2^ * _p_ *	*F*	*η* ^2^ * _p_ *	*F*	*η* ^2^ * _p_ *	(Post)
Left-hand grip Strength (kg)	22.956	0.167 ***	25.815	0.070 ***	0.949	0.008	con < CST, con < CST + GS, CST > GS, CST > CST + GS
Right-hand grip strength (kg)	31.888	0.218 ***	30.556	0.082 ***	1.180	0.010	con < CST, con < CST + GS, GS < CST + GS, GS < CST, CST > CST + GS
Long jump (cm)	4.041	0.035 **	6.100	0.018 *	0.083	0.001	con < CST, GS < CST
50 m (s)	9.500	0.079 ***	25.056	0.070 ***	2.240	0.020	con < CST, con < CST + GS, con < GS, GS < CST
800/1000 m (min)	10.181	0.084 ***	0.579	0.002	0.441	0.004	con < CST, con < CST + GS, GS < CST, GS < CST + GS
Core endurance (score)	21.046	0.157 ***	0.186	0.001	2.308	0.020	con < CST, con < CST + GS, con < GS, CST + GS > GS, CST + GS > CST, CST > GS

The *p*-value is the result of the covariance test, setting age, BMI, and baseline test as covariates and group and gender as fixed factors. “*”: *p* < 0.05; “**”: *p* < 0.01; “***”: *p* < 0.001.

**Table 6 ijerph-19-07715-t006:** Descriptive statistics of exercise attitude (mean, standard deviation); within-group analysis.

Variables	GS			CST			GS + CST			CON		
[Score]	Pre	Post	*d*	Pre	Post	*d*	Pre	Post	*d*	Pre	Post	*d*
Behavioral attitudes	30.2, 6.0	29.5, 6.2	0.13	29.1, 6.7	29.5, 7.0	0.05	32.8, 5.9	33.0, 6.0	5.96	31.0, 6.3	31.0, 6.0	0.04
Target attitudes	49.1, 6.6	47.5, 7.4	0.23 *	47.9, 8.6	47.8, 8.5	0.01	51.0, 7.4	50.8, 8.0	7.97	49.6, 8.0	48.5, 7.0	0.03
Behavioral perceptions	28.1, 3.9	28.3, 6.8	0.04	27.6, 4.9	27.8, 4.9	0.04	29.2, 4.5	28.9, 4.8	4.75	28.4, 4.6	28.9, 4.8	0.05
Behavioral habits	31.6, 8.8	30.2, 9.0	0.16 *	29.9, 9.0	30.4, 9.5	0.06	34.5, 9.0	35.1, 8.0	7.96	31.6, 8.0	31.8, 8.0	0.08
Behavioral intention	23.9, 6.8	23.1, 7.0	0.12	23.1, 7.4	23.9, 7.4	0.12	27.4, 7.3	27.7, 6.8	6.84	24.5, 6.8	24.5, 7.1	0.04
Emotional experience	32.9, 9.6	33.3, 8.4	0.04	33.1, 8.6	34.4, 9.2	0.14	37.4, 8.4	37.8, 8.2	8.20	34.8, 8.5	35.4, 8.7	0.05
Sense of Behavioral Control	24.2, 6.8	23.4, 6.2	0.12	24.0, 7.1	25.1, 7.5	0.15	26.5, 6.9	27.4, 7.0	6.95	25.7, 6.4	24.9, 7.3	0.13
Subjective standards	20.7, 5.7	20.3, 5.7	0.07	20.8, 4.6	21.3, 5.1	0.11	22.4, 5.3	21.8, 4.7	4.69	20.1, 4.9	21.0, 4.6	0.13

GS: Goal setting intervention group; CST: Core strength training intervention group; GS + CST: Goal setting and core strength training cointervention group; CON: Control group; statistical significance was set to *p* < 0.05; “*”: *p* < 0.05.

**Table 7 ijerph-19-07715-t007:** Analysis of covariance of exercise attitude test indexes between experimental and control groups.

Variables	Group		Gender		Interaction Effects		Pairwise Comparison Post
[Score]	*F*	*η* ^2^ * _p_ *	*F*	*η* ^2^ * _p_ *	*F*	*η* ^2^ * _p_ *	
Behavioral attitudes	2.620	0.022	3.327	0.010	0.192	0.002	—
Target attitudes	1.674	0.014	2.294	0.007	1.066	0.009	—
Behavioral perceptions	0.218	0.002	0.040	0.000	0.237	0.002	—
Behavioral habits	3.668	0.031 *	5.362	0.015 *	0.063	0.001	CST + GS > GS, CST + GS > CST
Behavioral intention	3.354	0.028 *	2.652	0.008	1.230	0.011	CST + GS > GS
Emotional experience	1.156	0.010	0.012	0.000	1.210	0.010	—
Sense of behavioral control	3.802	0.032 *	6.589	0.019 *	0.353	0.003	CST + GS > GS, CST > GS, CST + GS > con
Subjective standards	1.286	0.011	0.129	0.000	1.103	0.010	—

The *p*-value is the result of the covariance test, setting age, BMI, and baseline test as covariates and group and gender as fixed factors. “*”: *p* < 0.05.

## Data Availability

The datasets used and/or analyzed during the current study are available from the corresponding author upon reasonable request.

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
