# Peer review of "Effects of a SMART Goal Setting and 12-Week Core Strength Training Intervention on Physical Fitness and Exercise Attitudes in Adolescents: A Randomized Controlled Trial"

_ijerph, 2022, doi:10.3390/ijerph19137715_

Round 1

Reviewer 1 Report

Lots of great information. I appreciate being able to review the manuscript. I believe my comments are reasonable and helpful.

Line 10, 12-week

Line 23, it was not…

Line 23 and 26 are confusing about the exercise attitude result(s).

Effect size interpretations are needed in the abstract in addition to the p values.

Keywords, your keywords are in your title. You should change all of them.

Line 31, …are a global…

Line 79, exercise attitudes.

Line 87, did the adolescents get a say? Could they say no to the research?

2.2. Study design, it seems to me all this section should be in the introduction. It is great information. It reads as introduction material.

I do not read the Kyllo and Landers (1995) meta-analysis mentioned and references as important to the effectiveness of goal setting. - Goal Setting in Sport and Exercise: A Research Synthesis to Resolve the Controversy. Journal of Sport and Exercise Psychology. L. Blaine Kyllo & Daniel M. Landers

DOI: https://doi.org/10.1123/jsep.17.2.117

The tables and figures are informative. I do not see how to reduce them. They are a bit overwhelming.

I would like to see a table of effect size values. I see the partial eta squared. Those do not help us understand the differences in many of the figures.

Those are between two time points for a group or between two groups as opposed to the partial eta squared being for the entire ANOVA. I know you know this.

My point is for Figure 5 and 6 Cohen’s d or Hedges’ g above the *, **, or *** would be very informative. I could 31 places. Those calculations are simple to do given you have all the data.

Line 362, it seems again you should have this information for your data.

Figure 4 would be too overwhelming with within group Hedges’ g. I think the key is the between two groups Hedges’ g or Cohen’s d.

Table 7, the first partial eta is missing the first 0, 0.022.

Author Response

Dear reviewer,

Thank you for providing the detailed and constructive suggestions for our manuscript. We have carefully reviewed, addressed each comment below and made corresponding changes in the manuscript (marked up using the “Track Changes” function).

We hope you find the changes satisfactory.

Sincerely

Yijuan Lu

Comment 1: Line 10, 12-week

Response

Thank you for constructive suggestions.

Revision was made in the manuscript (line 10).

Comment 2: Line 23, it was not…

Response

Thank you for constructive suggestions.

We have made changes to the summary report and have removed this sentence.

Comment 3: Line 23 and 26 are confusing about the exercise attitude result(s).

Response

We apologize for the confusing description.

We have revised the abstract section and have removed this sentence.

Comment 4: Effect size interpretations are needed in the abstract in addition to the p values.

Response

Thank you for constructive suggestions.

Revision was made in the manuscript (line 17-27).

Comment 5: Keywords, your keywords are in your title. You should change all of them.

Response

Thank you for constructive suggestions.

Revision was made in the manuscript (line 29).

Comment 6: Line 31, …are a global…

Response

Thank you for constructive suggestions.

Revision was made in the manuscript (line 32).

Comment 7: Line 79, exercise attitudes.

Response

Thank you for constructive suggestions.

We have revised the introductory section and have removed this sentence.

Comment 8: Line 87, did the adolescents get a say? Could they say no to the research?

Response

We apologize for the confusing description.

Adolescents voluntarily chose to participate or not to participate in the experiment.

Revision was made in the manuscript (line 194-196).

Comment 9: 2.2. Study design, it seems to me all this section should be in the introduction. It is great information. It reads as introduction material.

Response

Thank you for constructive suggestions.

We moved parts of the study design to the introduction (line 94-155).

Comment 10: I do not read the Kyllo and Landers (1995) meta-analysis mentioned and references as important to the effectiveness of goal setting. - Goal Setting in Sport and Exercise: A Research Synthesis to Resolve the Controversy. Journal of Sport and Exercise Psychology. L. Blaine Kyllo & Daniel M. Landers

DOI: https://doi.org/10.1123/jsep.17.2.117

Response

We apologize for the confusing description.

The 36 articles included in this systematic review cover various categories of goal setting, such as difficult goals, easy goals, specific goals, vague/general goals, short-term goals, long-term goals. The effects of different categories of goal setting and other additional variables (e.g., goal setting reach, goal acceptance) varied. To improve the effectiveness of the intervention, we based the goal setting in the study on the SMART principle.

Comment 11: The tables and figures are informative. I do not see how to reduce them. They are a bit overwhelming.

Response

We apologize for the confusing description.

Figure 3 was removed from the revised manuscript and Tables 4-7 were optimized.

Comment 12: I would like to see a table of effect size values. I see the partial eta squared. Those do not help us understand the differences in many of the figures.

Response

We apologize for the confusing description.

Revision was made in the manuscript (Table 4-7 and Figures 5 & 6)

Comment 13: Those are between two time points for a group or between two groups as opposed to the partial eta squared being for the entire ANOVA. I know you know this.

Response

We apologize for the confusing description.

Revision was made in the manuscript (Table 4-7 and Figures 5 & 6)

Comment 14: My point is for Figure 5 and 6 Cohen’s d or Hedges’ g above the *, **, or *** would be very informative. I could 31 places. Those calculations are simple to do given you have all the data.

Response

Thank you for constructive suggestions.

Revision was made to add Cohen's d to the report in Figures 5 and 6.

Comment 15: Line 362, it seems again you should have this information for your data.

Response

Thank you for constructive suggestions.

Revision was made to add Cohen's d to the report.

Comment 16: Figure 4 would be too overwhelming with within group Hedges’ g. I think the key is the between two groups Hedges’ g or Cohen’s d.

Response

Thank you for constructive suggestions.

The revision of the manuscript was removed the p-value in Table 4 and added Cohen's d between the two groups.

Comment 17: Table 7, the first partial eta is missing the first 0, 0.022.

Response

Thank you for constructive suggestions.

Revision was made in the manuscript (Table 7)

Reviewer 2 Report

This is a well written manuscript that needs significantly more information presented.

Introduction:

1. In the first paragraph where you start out talking about how kids are less physically active and there is declining physical health in adolescents please provide some well cited statistics to back up that claim.

2. In the initial part of the introduction please provide physical activity (PA) recommendations for Chinese adolescents so that you can provide how these students are not meeting PA levels later (provide numbers)

3. I understand that you provide a reasoning for using TPB, but in your introduction you should introduce TPB and other potential methods and why TPB is the most appropriate.

4. In your final paragraph you should not describe what was done in the study, but rather what the objective of the study was and what the hypothesis was.

Methods:

The methods require significant more information

1. How were subjects recruited? Were all students given a permission slip and then only the students' who's parents signed the permission slip were included in the study? How many total students were recruited? What was the inclusion/exclusion criteria? 

2. How were students randomized into the different groups? Were they matched? Are the participants presented in Table 1 the ones who started the study or the ones that were left after the 2 who dropped out?

3. Which interventions did the 2 students drop out of?

4. Did you not include a warm-up for the 50m, grip strength and core test?

5. You state that you used a test that Brian McKenzie created. Can you please describe the test and how it is performed?

6. What was the order of testing for each testing day?

7. Did you check the data for normality? If so, how? Did you have to try to use any normalization efforts?

8. What were the confounders used as covariates? 

9. Did you conduct an a prior or post-hoc power analysis?

Results:

1. Were the results presented in Table 4 Bonferroni adjusted values?

2. Did you test for pre-intervention differences in the fitness scores?

3. From Table 4 you can remove the p-value and instead use ** (like you have done in Figure 4). Instead it would be better if you reported effect sizes

4.  For Table 5, please provide directionality of differences (i.e. con< GS, GS+CST, CST). You can remove the p-value and replace with effect sizes

5. For between group, what sort of post-hoc analysis did you use?

6. Please present the sex differences results in your results. Were there any group by sex differences in pre-tests at baseline?

7. I think there's a typo on line 246

8. In Figure 5, what are you presenting? Is that post-test scores? Why choose post-test scores instead of mean differences? It seems that there may be statistically significant differences in baseline scores on many of these tests.

9. Same comments apply to Figure 6

Discussion

I am having a tough time determining whether the discussion would be fully justified based on the lack of information in the methodology and results as that may dictate how some of the results are presented and interpreted. 

Author Response

Dear reviewer,

Thank you for providing the detailed and constructive suggestions for our manuscript. We have carefully reviewed, addressed each comment below and made corresponding changes in the manuscript (marked up using the “Track Changes” function).

We hope you find the changes satisfactory.

Sincerely

Yijuan Lu

Introduction:

Comment 1: In the first paragraph where you start out talking about how kids are less physically active and there is declining physical health in adolescents, please provide some well cited statistics to back up that claim.

Response

Thank you for constructive suggestions.

Revision was made in the manuscript (line 32-39).

  The physical inactivity and declining physical health of adolescents have become are a global health problem. Studies have shown that no more than 45% of students met the recommended level of 60 min of moderate and vigorous physical activity (MVPA) [1, 2], with only 8% of the adolescent population meeting the standard [3]. In Finland, 90% of girls and 77% of boys did not meet the daily recommended physical activity in their self-report [4]. In several European countries, accelerometer measurements in children aged 10–12 years showed that only 4.6% of girls as well as 16.8% of boys met the recommended amount [5].

Comment 2: In the initial part of the introduction please provide physical activity (PA) recommendations for Chinese adolescents so that you can provide how these students are not meeting PA levels later (provide numbers).

Response

Thank you for constructive suggestions.

Revision was made in the manuscript (line 40-47).

  According to nationwide surveys, the level of physical activity and health status of Chinese school-aged youths showed a downward trend from around 22.7% in 2010 to only about 8.9% in 2014 [6]. Although many policies have been put in place to improve adolescents’ PA level, less than one-third of Chinese adolescents could meet the recommended level of 60 min of MVPA in 2016 [6]. As the physical activity level of teenagers can continue to decrease with age, which could be well above the recommended MVPA at 9 years but only 17% by 15 years [7], it is very important to conduct effective interventions to improve the physical activity of Chinese adolescents.

Comment 3: I understand that you provide a reasoning for using TPB, but in your introduction you should introduce TPB and other potential methods and why TPB is the most appropriate.

Response

Thank you for constructive suggestions.

Revision was made in the manuscript (line 94-106).

  The theory of planned behavior (TPB) proposed by Ajzen [8] can help us understand how people change their behavior patterns. This model suggests that the most important determinant of an individual’s behavior is their intention to perform that behavior, with three cognitive variables—attitudes, subjective norms, and perceived behavioral control—which are said to be the direct determinants of intention. Attitudes represent a key explanatory variable in many theories of health behavior; research has shown that attitudes predict both intention and behavior [9]. The TPB provides a useful conceptual framework for marking a sense of the complexities of human social behavior [10]. The attitude toward physical activity is a combination of individuals’ cognitive evaluations, affective experiences, and behavioral intentions related to physical activity [11], which are all essential psychological factors for individuals to persist in physical activity, and adolescents’ positive attitudes toward physical activity have positive impacts on their physical fitness [12].

Comment 4: In your final paragraph you should not describe what was done in the study, but rather what the objective of the study was and what the hypothesis was.

Response

Thank you for constructive suggestions.

Revision was made in the manuscript (line 156-159).

  Based on the above, this study (1) proposes a comprehensive intervention theoretical framework based on TPB and GST, (2) designs an intervention program based on the comprehensive intervention theoretical framework, and (3) aims to evaluate the effect of the intervention on enhancing adolescents' exercise attitudes and physical fitness.

Methods:

Comment 1: How were subjects recruited? Were all students given a permission slip and then only the students' whose parents signed the permission slip were included in the study? How many total students were recruited? What was the inclusion/exclusion criteria?

Response

We apologize for the confusing description.

Revision was made in the manuscript (line 194-217).

    Three-hundred-and-sixty-eight students were chosen from eight classes in two middle schools in Hangzhou, Zhejiang Province, China, by random sampling, based on their voluntary participation in this study. The inclusion criteria require that (a) students are able to perform daily physical education classes, and (b) students do not participate in the school's after-school sports club. Four students were excluded because they participated in daily training with the school's soccer club. Of the 368 candidates, 364 volunteers met the inclusion criteria, as described in Figure 3. Before the study, the demographic information of these students was collected, then we allocated them to the intervention (GS, CST, GS + CST) and control groups. A total of 364 students were pretested at baseline, and two students (one in the GS + CST group due to a sprained ankle; one in the control group due to transfer to another school) were lost to follow–up at the posttest. All the participants completed the psychological questionnaire.

    Hence, a total of 271 students of the intervention group (138 boys and 133 girls) and 91 subjects of the control group (44 boys and 47 girls) were analyzed via screening and cleaning data for questionnaires and physical fitness test (shown in Table 1). The students’ mean ± SD age and body mass index (BMI) were 14.5 ± 1.07 years and 19.82 ± 3.64, respectively. All the students and families were fully informed about the possible problems related to the experimental procedures. The study procedures were approved by the research ethics board of Zhejiang University (No.2020-002, 2020.07.22). All the participants gave written informed consent. The analyses were processed using the SPSS 25.0 program, except for effect size calculations (Cohen's d), which were processed using a web-based effect-size calculator . This calculator is a companion to the 2001 book by Mark W. Lipsey and David B. Wilson, Practical Meta-analysis, published by Sage.

Comment 2: How were students randomized into the different groups? Were they matched? Are the participants presented in Table 1 the ones who started the study or the ones that were left after the 2 who dropped out?

Response

We apologize for the confusing description.

The participants in Table 1 are those left after 2 dropped out.

Revision was made in the manuscript (line 175-178, line 206-208, Figure 3).

  This study was designed as a randomized controlled trial. Before the trial was initiated, the participants were randomly assigned to one of the four groups using a computer-generated simple randomization procedure [13].

  Hence, a total of 271 students of the intervention group (138 boys and 133 girls) and 91 subjects of the control group (44 boys and 47 girls) were analyzed via screening and cleaning data for questionnaires and physical fitness test (shown in Table 1).

Comment 3: Which interventions did the 2 students drop out of?

Response

We apologize for the confusing description.

Revision was made in the manuscript (Figure 3, line 202-204).

A total of 364 students were pretested at baseline, and two students (one in the GS + CST group due to a sprained ankle; one in the control group due to transfer to another school) were lost to follow–up at the posttest.

Comment 4: Did you not include a warm-up for the 50m, grip strength and core test?

Response

We apologize for the confusing description.

Revision was made in the manuscript (line 256-263).

With the assistance of the experiential schools, the physical fitness data were tested during the first week before the intervention, in strict accordance with the requirements of the Chinese National Student Physical Fitness Test. Physical fitness tests were completed by three trained experimenters assisting the physical education teachers. All tests (5 items) were completed in one physical education classroom session. Before the test, students complete a 5-minute warm-up exercise led by the physical education teacher. The tests were arranged in the order of grip strength, long jump, 50-meter dash, core endurance test, and 800-/1000-meter run.

Comment 5: You state that you used a test that Brian McKenzie created. Can you please describe the test and how it is performed?

Response

We apologize for the confusing description.

Revision was made in the manuscript (line 298-322).

Three trained experimenters accompanying the physical education teachers completed the core endurance test. Prior to the test, each tester was assigned a pen, a physical fitness test sheet (including core strength) and a clock. The testers will demonstrate and explain the eight test movements and requirements. The content and order of the tests are as follows.

  1. Participants will start in a planking position and hold for 60 seconds.

(1 point for completion)

  1. Lift their right arm off the ground and hold for 15 seconds.

(3 point for completion)

  • Return their right arm to the ground and lift the left arm off the ground and hold for 15 seconds.

(5 point for completion)

  1. Return their left arm to the ground and lift the right leg off the ground hold for 15 seconds.

(6 point for completion)

  1. Return their right leg to the ground and lift the left leg off the ground and hold for 15 seconds.

(10 point for completion)

  1. Lift their left leg and right arm off the ground hold for 15 seconds.

(15 point for completion)

  • Return their left leg and right arm to the ground. Lift your right leg and left arm off the ground hold for 15 seconds.

(25 point for completion)

  • Return to the plank exercise position (elbows on the ground) and hold this position for 30 seconds.

(35 point for completion)

It is important to note that the trunk is always in a neutral position throughout the test, and the test must be performed continuously from the first step to the eighth step, if a step in the test fails to meet the requirements, the test is over, and the total score obtained at this point is the test result. The higher the score, the better the core strength and stability.

Comment 6: What was the order of testing for each testing day?

Response

We apologize for the confusing description.

Revision was made in the manuscript (line 256-263, line 326-327).

Physical Fitness Test: With the assistance of the experiential schools, the physical fitness data were tested during the first week before the intervention, in strict accordance with the requirements of the Chinese National Student Physical Fitness Test. Physical fitness tests were completed by three trained experimenters assisting the physical education teachers. All tests (5 items) were completed in one physical education classroom session. Before the test, students complete a 5-minute warm-up exercise led by the physical education teacher. The tests were arranged in the order of grip strength, long jump, 50-meter dash, core endurance test, and 800-/1000-meter run.

Physical Attitude Test: With the assistance of a physical education teacher, the researcher collected data on participants' attitudes toward exercise during the first week before the intervention.

Comment 7: Did you check the data for normality? If so, how? Did you have to try to use any normalization efforts?

Response

We apologize for the confusing description.

Revision was made in the manuscript (line 344-348).

The assumptions of the ANOVA were assessed to be satisfied based on the results of the Shapiro-Wilk and Levene tests. When the data were not normally distributed, the Mann–Whitney U test was performed for between-groups comparisons and the Wilcoxon matched-pair test was used for within-groups comparisons. According to the Shapiro–Wilk and the Levene test results, ANCOVA assumptions were met.

Comment 8: What were the confounders used as covariates?

Response

We apologize for the confusing description.

Revision was made in the manuscript (line 353-355).

A factorial univariate analysis of covariance (ANCOVA) utilizing the baseline score and other key confounders as covariates (age and BMI) was used to determine the effects of intervention.

Comment 9: Did you conduct an a prior or post-hoc power analysis?

Response

We apologize for the confusing description.

We conducted a post hoc power analysis.

Revision was made in the manuscript (line 358-360).

Analyses of simple effects and post hoc Bonferroni adjustments were performed after significant interaction effects by overall ANCOVA were confirmed.

Results:

Comment 1: Were the results presented in Table 4 Bonferroni adjusted values?

Response

We apologize for the confusing description.

The results in Table 1 are Bonferroni-adjusted.

Comment 2: Did you test for pre-intervention differences in the fitness scores?

Response

We apologize for the confusing description.

We tested for differences in pre-intervention fitness scores, but did not report them in the manuscript. Additions were made in the revision.

Revision was made in the manuscript (line 367-369, line 440-442).

An ANOVA was used to test for differences between groups for the baseline test. The results showed that except for the significant difference in right-hand grip strength (p = 0.03), the differences in the rest of the physical fitness tests were not significant.

ANOVA was used to test for between-group differences in the baseline tests. The results showed that there were significant differences in the exercise attitude component, except for target attitudes, and behavioral perceptions (p > 0.05).

Comment 3: From Table 4 you can remove the p-value and instead use ** (like you have done in Figure 4). Instead, it would be better if you reported effect sizes

Response

Thank you for constructive suggestions.

Revision was made in the manuscript (Table 4, 6).

Comment 4: For Table 5, please provide directionality of differences (i.e., con< GS, GS+CST, CST). You can remove the p-value and replace with effect sizes

Response

Thank you for constructive suggestions.

Revision was made in the manuscript (Table 5, 7).

Comment 5: For between group, what sort of post-hoc analysis did you use?

Response

We apologize for the confusing description.

Revision was made in the manuscript (Line 461, 403).

Variables with significant differences were compared Bonferroni post hoc to further understand the differences between the four groups.

Comment 6: Please present the sex differences results in your results. Were there any group by sex differences in pre-tests at baseline?

Response

We apologize for the confusing description.

Revision was made in the manuscript (line 369-372).

Gender differences between groups were analyzed using the chi-square test, and p-values were corrected using the Bonferroni method. The results showed that the gender differences between the four groups were not significant (c2 = 0.273, p = 0.965).

Comment 7: I think there's a typo on line 246

Response

We apologize for the confusing description.

Revision was made in the manuscript.

Comment 8: In Figure 5, what are you presenting? Is that post-test scores? Why choose post-test scores instead of mean differences? It seems that there may be statistically significant differences in baseline scores on many of these tests.

Response

We apologize for the confusing description.

Figure 5 shows the post-test scores.

In our study, a factorial univariate analysis of covariance (ANCOVA) utilizing the baseline score and other key confounders as covariates (age and BMI) was used to determine the effects of post-intervention. In addition, there were no statistically significant differences in baseline physical fitness scores between the groups.

Comment 9: Same comments apply to Figure 6

Response

We apologize for the confusing description.

In our study, a factorial univariate analysis of covariance (ANCOVA) utilizing the baseline score and other key confounders as covariates (age and BMI) was used to determine the effects of post-intervention.

Discussion

Comment 1: I am having a tough time determining whether the discussion would be fully justified based on the lack of information in the methodology and results as that may dictate how some of the results are presented and interpreted.

Response

We apologize for the confusing description.

We have added methods and results to the manuscript in accordance with your revised comments, which we hope will further help you understand the discussion section.

Reviewer 3 Report

This paper is a clinical trial, the type of study that provides the most scientific evidence.

1.       Please note  that MDPI Instructions for authors  the editorial adhere to the CONSORT statement in clinical trials 

https://www.mdpi.com/journal/information/instructions

CONSORT Statement

MDPI requires a completed CONSORT 2010 checklist and flow diagram as a condition of submission when reporting the results of a randomized trial. Templates for these can be found here or on the CONSORT website (http://www.consort-statement.org) which also describes several CONSORT checklist extensions for different designs and types of data beyond two group parallel trials. At minimum, your article should report the content addressed by each item of the checklist.

It is imperative to describe the randomization process.

2.       The first time the TPB and GST are cited in the introduction, the corresponding bibliographic references should be introduced. Right now, they are in the Material and Methods section.

3.       Concerning the PE Midterm, could the authors give some more details, what factors it measures, and is it done by the school or externally. Authors must understand that the reader may not be familiar with the Chinese education system.

4.       Explain the acronym SMART

5.       In the abstract, there is much emphasis on the Ps. Although the P is essential, the magnitude of the effect should be indicated. For example, one could compare two diets and see that one helps lose more weight than the other (P < 0.001), but the weight loss could be only 1g. Therefore, the magnitude should be stated.

6.       In the text of the results, the differences should be emphasized. It is not enough to say that there were differences in 50m in groups X, Y, and Z (P < 0.001), the magnitude of the differences should be commented on.

7.       It is indicated that two students dropped out of the study. Can the authors provide some information on the reasons? Also, indicate in which groups the students were assigned.

8.       Review the bibliographic references, in reference no. 47 the publisher and city of publication are missing. In reference 39, the volume and pages of the journal article are missing.

Author Response

Dear reviewer,

Thank you for providing the detailed and constructive suggestions for our manuscript. We have carefully reviewed, addressed each comment below and made corresponding changes in the manuscript (marked up using the “Track Changes” function).

We hope you find the changes satisfactory.

Sincerely

Yijuan Lu

Comments 1: Please note that MDPI Instructions for authors the editorial adhere to the CONSORT statement in clinical trials. MDPI requires a completed CONSORT 2010 checklist and flow diagram as a condition of submission when reporting the results of a randomized trial. Templates for these can be found here or on the CONSORT website (http://www.consort-statement.org) which also describes several CONSORT checklist extensions for different designs and types of data beyond two group parallel trials. At minimum, your article should report the content addressed by each item of the checklist. It is imperative to describe the randomization process.

Response

Thank you for the constructive suggestions.

Revision was made in the manuscript (Figure 3, line 178-182).

This RCT was reported according to the Consolidated Standard of Reporting Trials (CONSORT 2010) guidelines (http://www.consort-statement.org; accessed on 15 June 2022). In addition, a concise overview of the intervention programs was described according to the CONSORT 2010 checklist (http://www.consort-statement.org; accessed on 15 June 2022).

Comments 2: The first time the TPB and GST are cited in the introduction, the corresponding bibliographic references should be introduced. Right now, they are in the Material and Methods section.

Response

Thank you for constructive suggestions.

Revision was made in the manuscript (line 94-137).

Comments 3: Concerning the PE Midterm, could the authors give some more details, what factors it measures, and is it done by the school or externally. Authors must understand that the reader may not be familiar with the Chinese education system.

Response

We apologize for the confusing description.

Revision was made in the manuscript (line 75-93).

Comments 4: Explain the acronym SMART

Response

We apologize for the confusing description.

Revision was made in the manuscript (Line 126-137).

The SMART principle consists of five components of effective goals: They must be specific, measurable, attainable, relevant, and timely.

S stands for special, which means that the goal setting or performance evaluation criteria must be specific, so that people know what to do.

M stands for measurable, which means that the goal or target should be measurable and able to give clear judgment, such as through data.

A stands for Attainable, which means that when setting goals for yourself or others, the goals should not be too high or too low, if they are too high, it is easy to discourage people, if they are too low and unchallenging, it is better to work hard to reach them.

R stands for relevant, which means that there should be some relevance between the goal and the target, and the whole is for the big goal or the big direction.

T represents time bound, that is, the deadline, for a goal, if there is no deadline, then it is basically the same as invalid, which is the biggest enemy of procrastination.

Comments 5: In the abstract, there is much emphasis on the Ps. Although the P is essential, the magnitude of the effect should be indicated. For example, one could compare two diets and see that one helps lose more weight than the other (P < 0.001), but the weight loss could be only 1g. Therefore, the magnitude should be stated.

Response

We apologize for the confusing description.

Revision was made in the manuscript (line 17-27).

We have added reporting of effect size in the abstract.

The differences in post hoc comparisons are reported in the results.

Comments 6: In the text of the results, the differences should be emphasized. It is not enough to say that there were differences in 50m in groups X, Y, and Z (P < 0.001), the magnitude of the differences should be commented on.

Response

We apologize for the confusing description.

Revision was made in the manuscript (line 403-428, line 463-471).

Comments 7: It is indicated that two students dropped out of the study. Can the authors provide some information on the reasons? Also, indicate in which groups the students were assigned.

Response

We apologize for the confusing description.

Revision was made in the manuscript (Figure 3, line 202-204).

A total of 364 students were pretested at baseline, and two students (one in the GS + CST group due to a sprained ankle; one in the control group due to transfer to another school) were lost to follow–up at the posttest.

Comments 8: Review the bibliographic references, in reference no. 47 the publisher and city of publication are missing. In reference 39, the volume and pages of the journal article are missing.

Response:

Thank you for constructive suggestions.

Revision was made in references of the manuscript.

Round 2

Reviewer 2 Report

I'd like to thank the authors for making these major revisions. The added information makes the manuscript significantly better.